# Captopril, a Renin–Angiotensin System Inhibitor, Attenuates Tumour Progression in the Regenerating Liver Following Partial Hepatectomy

**DOI:** 10.3390/ijms23095281

**Published:** 2022-05-09

**Authors:** Georgina E. Riddiough, Katrina A. Walsh, Theodora Fifis, Georgios Kastrappis, Bang M. Tran, Elizabeth Vincan, Vijayaragavan Muralidharan, Christopher Christophi, Claire L. Gordon, Marcos V. Perini

**Affiliations:** 1Department of Surgery, The University of Melbourne, Austin Health, Lance Townsend Building, Level 8, 145 Studley Road, Heidelberg, VIC 3084, Australia; georgina.riddiough@unimelb.edu.au (G.E.R.); kawalsh@unimelb.edu.au (K.A.W.); tfifis@unimelb.edu.au (T.F.); glk@student.unimelb.edu.au (G.K.); v.muralidharan@unimelb.edu.au (V.M.); c.christophi@unimelb.edu.au (C.C.); 2Department of Infectious Diseases, The Peter Doherty Institute for Infection and Immunity, The University of Melbourne, Melbourne, VIC 3000, Australia; manht@unimelb.edu.au (B.M.T.); evincan@unimelb.edu.au (E.V.); 3Victorian Infectious Disease Reference Laboratory, The Peter Doherty Institute for Infection and Immunity, Melbourne, VIC 3000, Australia; 4Curtin Medical School, Curtin University, Perth, WA 6102, Australia; 5Department of Infectious Diseases, Austin Health, 145 Studley Road, Heidelberg, VIC 3084, Australia; claire.gordon@austin.org.au; 6Department of Microbiology & Immunology, The Peter Doherty Institute for Infection and Immunity, The University of Melbourne, Melbourne, VIC 3000, Australia; 7North Eastern Public Health Unit, Austin Health, 145 Studley Road, Heidelberg, VIC 3084, Australia

**Keywords:** surgical oncology, liver regeneration, immunology, hepatic tissue-resident memory T cells, liver neoplasms, neoplasm metastasis

## Abstract

(1) Liver regeneration following partial hepatectomy for colorectal liver metastasis (CRLM) has been linked to tumour recurrence. Inhibition of the renin–angiotensin system (RASi) attenuates CRLM growth in the non-regenerating liver. This study investigates whether RASi exerts an antitumour effect within the regenerating liver following partial hepatectomy for CRLM and examines RASi-induced changes in the tumour immune microenvironment; (2) CRLM in mice was induced via intrasplenic injection of mouse colorectal tumour cells, followed by splenectomy on Day 0. Mice were treated with RASi captopril (250 mg/kg/day), or saline (control) from Day 4 to Day 16 (endpoint) and underwent 70% partial hepatectomy on Day 7. Liver and tumour samples were characterised by flow cytometry and immunofluorescence; (3) captopril treatment reduced tumour burden in mice following partial hepatectomy (*p* < 0.01). Captopril treatment reduced populations of myeloid-derived suppressor cells (MDSCs) (CD11b^+^Ly6C^Hi^ *p* < 0.05, CD11b^+^Ly6C^Lo^ *p* < 0.01) and increased PD-1 expression on infiltrating hepatic tissue-resident memory (T_RM_)-like CD8^+^ (*p* < 0.001) and double-negative (CD4^-^CD8^-^; *p* < 0.001) T cells; (4) RASi reduced CRLM growth in the regenerating liver and altered immune cell composition by reducing populations of immunosuppressive MDSCs and boosting populations of PD-1^+^ hepatic T_RM_s. Thus, RASi should be explored as an adjunct therapy for patients undergoing partial hepatectomy for CRLM.

## 1. Introduction

Liver resection offers the best chance of cure for patients with colorectal liver metastasis (CRLM); however, tumour recurrence in the future liver remnant (FLR) is not infrequent and affects 40% of patients [1,2]. The biological processes and signalling pathways underlying physiological liver regeneration are implicated in tumour recurrence following liver resection [3]. Most liver regeneration occurs within two weeks of surgery and complete restoration of hepatic mass is achieved within 3 months in humans and 10 days in mice [4]. Neoadjuvant and adjuvant chemotherapy has been shown to improve patient survival after CRLM resection [5,6]; however, chemotherapeutic agents are avoided during the immediate perioperative period due to hepatotoxicity and impairment of early postoperative liver regeneration. Novel therapies which safely promote antitumour responses during the immediate postoperative period are, therefore, required.

Renin–angiotensin system inhibitors (RASis), commonly used to treat cardiac failure and hypertension, have been shown to prolong survival, improve response to neoadjuvant therapy, and reduce recurrence in a range of solid cancers [7,8,9]. Studies have reported that RASis exhibit a range of antitumour activities, including immunomodulatory, antiproliferative, and antiangiogenic effects [3,10,11]. Additionally, we recently reported that one type of RASi, captopril, exhibits its antitumour effects in the regenerating liver via *c-myc* and *cyclin D1* downregulation [12]. In the non-regenerating liver, it has been shown that RASi therapy reprograms the adaptive antitumour immune response by enhancing populations of CD8^+^ cytotoxic and CD4^-^CD8^-^ (double-negative (DN)) T cells [13]; however, it is not known if these changes are also observed in the regenerating liver. To address this question, this study investigates the effect of captopril in the regenerating liver, on both innate and adaptive immune responses, using a model of CRLM and partial hepatectomy, which closely mimics the clinical scenario in which patients undergo liver resection for CRLM when clinically undetectable micrometastases may be present in the FLR.

## 2. Results

### 2.1. Captopril Treatment Reduces CRLM Tumour Burden in the Regenerating Liver

Captopril treatment significantly reduced CRLM tumour burden in the FLR following partial hepatectomy, compared with the control group at Day 16 (Figure 1a,b). This demonstrates that captopril is capable of inhibiting tumour growth within the milieu of the regenerating liver following partial hepatectomy.

### 2.2. Captopril Treatment Significantly Increases PD-1 Expression in T Cells

We recently reported that the RASi captopril induces changes in T-lymphocyte subpopulations and their activation status in a non-regenerating model of CRLM [13]. The present study was specifically focused on the immunomodulatory changes observed in the regenerating liver, the situation most relevant following liver resection.

Changes observed in the regenerating liver were complimentary to those previously observed in the non-regenerating liver [13]. Captopril treatment significantly reduced the ratio of CD4^+^:CD8^+^ (control 1.57 vs. captopril 0.91, *p* = 0.01) (Table 1) and CD4^+^:DN T (control 1.63 vs. captopril 0.87, *p* = 0.001) cells in liver (Appendix A).

To gain insight into the activation status of T lymphocytes, we examined the immune regulatory marker, PD-1. Treatment with captopril significantly increased the proportion of CD8^+^ and double-negative (DN; CD4^-^CD8^-^) T-cell populations expressing PD-1, and the intensity of PD-1 expression, in both regenerating liver and tumour specimens (Figure 1e,f). Immunofluorescence staining demonstrated that PD-1^+^ CD8^+^ T cells are located predominantly within tumour tissue (Figure 1g). These data suggest that captopril induces a phenotypic shift towards the expression of the activation marker and immune regulator, PD-1, on CD8^+^ and DN T cells.

### 2.3. Captopril Treatment Enhances Populations of T_RM_-like Cells and Increases PD-1 Expression on CD8^+^ T_RM_-Like Cells

Non-circulating T cells, termed tissue-resident memory T cells (T_RM_) [14], are increasingly implicated in mediating effective and sustained antitumour responses in solid cancers such as lung [15] and breast cancer [16]. We investigated whether the T-cell populations observed in regenerating liver and tumour tissue had features of T_RM_, defined by the expression of murine hepatic T_RM_ markers, CD44 and CD69 (termed “T_RM_-like cells”), and assessed the effect of captopril on these populations and their PD-1 expression. We found that, although the proportion of T_RM_-like CD8^+^ T cells did not increase with captopril treatment in the liver (Figure 2a), the ratio of CD4^+^ T_RM_-like:CD8^+^ T_RM_-like cells significantly reduced (control 1.00 vs. captopril 0.45, *p* = 0.01, Table 1). PD-1 expression on CD8^+^ T_RM_-like T cells significantly increased (Figure 2d–f). Conversely, the proportion of CD4^+^ T_RM_-like cells decreased in the regenerating liver following captopril treatment (Figure 2a), while their PD-1 expression was unchanged (Figure 2c,e,f).

### 2.4. Captopril Treatment Significantly Enhances Populations of T_RM_-Like DN T Lymphocytes and Increases PD-1 Expression on T_RM_-Like DN T Cells

The proportion of DN T cells co-expressing CD44 and CD69 (Figure 3a, “DN T_RM_-like”) in captopril-treated regenerating liver tissue significantly increased, and this corresponded with an increase in PD-1 expression on DN T_RM_-like cells in captopril-treated regenerating liver and tumour (Figure 3c,d).

### 2.5. In the Absence of Tumour, Captopril Treatment Enhances Populations of PD-1 Expressing T_RM_-Like CD8^+^ and DN T Cells

As tumour induces many changes in immune cell infiltrates and responses, we sought to extract the effect of captopril on tumour-related changes by comparing them with those changes that occurred in response to liver regeneration alone. In the absence of a tumour, captopril significantly increased the proportion of CD8^+^ T lymphocytes (Figure 4a; 37.0% vs. 59.0%, control vs. treatment) while reducing the proportion of CD4^+^ and DN T lymphocytes (Figure 4a). Additionally, PD-1^+^ expression significantly increased in CD8^+^ T lymphocytes (Figure 4b). Furthermore, captopril treatment increased the proportions of both CD8^+^ T_RM_-like cells (Figure 4c) and PD1^+^CD8^+^ T_RM_-like cells (Figure 4d) in regenerating liver without a tumour. The increase in CD8^+^ T_RM_-like cells with captopril treatment was more substantial in the absence of tumour (Figure 4c; 15.8% vs. 41.9%, control vs. treatment, *p* < 0.001), compared with that observed in the CLRM model (Figure 2a; 10.3% vs. 18.3%, *p* = 0.2) and this was also accompanied by a large reduction in the CD4^+^ T_RM_-like:CD8^+^ T_RM_-like ratio (control 0.47 vs. captopril 0.13, *p* < 0.001, Appendix A). Additionally, we found that captopril reduced the population of T_RM_-like DN T cells (Figure 4c; 9.5% vs. 5.3%, control vs. treatment, *p* < 0.05), compared with the result seen in the presence of a tumour in which case this population was significantly increased (Figure 3a; 7.1% vs. 10.7%, control vs. treatment, *p* < 0.05). Like the CRLM model, we observed that captopril treatment increased PD-1 expression on both CD8^+^ and DN T T_RM_-like cells (Figure 4b). These data suggest that captopril boosts the proportion of T_RM_-like CD8^+^ T cells (but not DN T cells) and increases the expression of PD-1 in CD8^+^ and DN T_RM_-like populations in physiological liver regeneration. This suggests that the expansion of DN and PD-1^+^ DN T_RM_-like populations is a tumour-specific response.

### 2.6. Captopril Treatment Modulates Myeloid-Derived Suppressor Cell Populations

Myeloid-derived suppressor cells (MDSCs) have been recognised to have immunosuppressive effects and are associated with cancer progression and poorer prognosis [17].

In the mouse model with CRLM, captopril treatment was associated with an overall reduction in MDSC populations, demonstrated by a reduction in CD11b^+^ leukocytes in the liver (Figure 5a). MDSC populations were distinguished into three ontological phenotypes based on their expression of Ly6C; CD11b+ Ly6C^Hi^ (monocytic MDSCs), Ly6C^Int^ (intermediate), and Ly6C^Lo^ (granulocytic MDSCs) [18,19]. Captopril treatment was associated with a reduction in both monocytic (Ly6C^Hi^) and granulocytic (Ly6C^Lo^) MDSC populations (Figure 5a).

Conversely, in the mouse model without CRLM, captopril did not modulate MDSC populations; no difference was found in either the total CD11b^+^ MDSCs or the MDSC subtypes (Figure 5b). Taken together, these data suggest that the RASi captopril may improve the overall antitumour immune response in the regenerating liver by reducing populations of MDSCs.

We have previously shown that captopril treatment modulates macrophage levels in CRLM, within the non-regenerating liver [20]. To investigate whether captopril modulated macrophage populations within the regenerating liver, we examined the expression of F4/80 on CD11b^+^Ly6C populations. Captopril treatment did not alter the expression of macrophage marker F4/80 on CD11b^+^Ly6C populations in either the regenerating model with or without CRLM (Appendix A).

PDL1 is an inhibitory ligand most commonly expressed on MDSCs including F4/80^+^ macrophages and tumour cells. Captopril treatment was associated with an increase in PDL1 expression on Ly6C^Hi^F4/80^+^ and Ly6C^Int^F4/80^+^ leukocytes in the regenerating livers in the model with CRLM (Appendix A). However, this response was not observed in the model without CRLM (Appendix A).

Notably, populations of MDSCs were higher in the presence of CRLM, compared with those in the absence of CRLM (control mean with tumour, total CD11b^+^ = 46% vs. control mean without tumour, total CD11b^+^ = 33%) (Figure 5a,b). Interestingly, captopril treatment reduced these populations in the presence of tumour (captopril mean total CD11b^+^ = 34%, Figure 5a), back down to the baseline level observed in the regenerating liver in the absence of CRLM (Figure 5b). Taken together, these findings highlight that, although the MDSCs present in the tissue respond by upregulating PDL1 expression, the sum effect of captopril is to reduce the presence of MDSCs.

## 3. Discussion

Tumour recurrence stimulated by the process of liver regeneration, following liver resection, is a major barrier to the long-term survival of patients with primary or secondary liver tumours. Presently, 40% of patients undergoing liver resection will experience intrahepatic recurrence. In this study, using a homogenous tumour mouse model that closely mimics the clinical scenario, we showed that RASi therapy with captopril inhibits the growth of CRLM in the regenerating FLR and is associated with phenotypic changes in immune cells in the tumour microenvironment. These changes included a reduction in populations of MDSCs and enhanced populations of PD-1^+^ hepatic-T_RM_-like T lymphocytes. Further research is required to determine whether the changes in the immune microenvironment induced by captopril therapy directly reduce tumour burden in the regenerating liver and how these immune changes relate to other known antitumour effects of captopril [3,10,11]. Nevertheless, in the future, perioperative administration of a RASi could be an effective therapy for reducing micrometastatic tumour growth in the FLR and improving clinical outcomes following liver resection for CRLM.

Studies have demonstrated that T_RM_s are involved in maintaining effective antitumour immune responses and correlate positively with survival for a range of solid cancers [15,16]. In contrast to other cancers, for example, melanoma and renal cell carcinoma, where inhibition of the PD-1-PDL1 pathway using checkpoint inhibitors has improved clinical outcomes by reinvigorating antitumour T cell responses [21,22], T_RM_ expressing PD-1 have been shown to maintain cytotoxicity in the tumour microenvironment (TME) [15,16,23,24]. This challenges the assumption that cytotoxic T cells expressing checkpoint markers such as PD-1 are tolerogenic or exhausted, which may explain why responsiveness to checkpoint inhibitors has not been universally observed in all cancer types, particularly if cancers rely on T_RM_ for antitumour activity, or there is an abundance of MDSCs. These findings correlate with those revealed in our study, in which a significant reduction in tumour burden was observed alongside the expansion of PD-1^+^ T_RM_ populations. Our findings support a role for PD1^+^ CD8^+^ T_RM_-like lymphocytes in maintaining cytotoxicity against CRLM. This may explain why some cancers, including mismatch repair proficient colorectal cancers, have exhibited poor responses to checkpoint inhibitor therapy and suggests that RASi could play a role in the treatment of CRC by modulating the immune infiltrate.

MDSCs are crucial components of TME that facilitate cancer growth by secreting immunosuppressive cytokines such as TNF and TGFβ, inadvertently allowing cancer cells to evade the patient’s immune system [25,26]. Clinical studies have shown that an abundance of tumour-infiltrating MDSCs is associated with a poorer prognosis [17]. Both granulocytic and monocytic MDSCs possess immunosuppressive properties such as inhibiting effector T cells and promoting the inhibitory effects of regulatory T cells [27]. These findings support the data presented here, which show that tumour burden is reduced alongside a significant reduction in granulocytic (Ly6C^Lo^) and monocytic (Ly6C^Hi^) MDSCs.

In acute inflammation, the adaptive and innate arms of the immune system work in concert with one another to restore homeostasis. In contrast, during chronic inflammation, such as within the TME, the immune response is dysregulated and characterised by the accumulation of immunoinhibitory cells, including MDSCs. Therefore, the finding we report here—namely, that certain T_RM_-cell populations are enhanced, while MDSC populations are reduced—is not likely to be coincidental. Studies in mice have demonstrated that both granulocytic and monocytic MDSCs suppress cytotoxic T cells [28]. However, further experiments are required to confirm that the MDSC changes observed here are directly responsible for the expansion of T_RM_ populations. To date, immunotherapy has largely focused on improving the cytotoxic function of effector T cells, rather than reducing populations of MDSCs, but the latter is also emerging as a viable approach to improve clinical cancer outcomes and trials examining this are underway [29]. Clinical trials examining the outcomes of treatment with anti-PD-1 therapies in CRC have only demonstrated benefits for patients with mismatch repair deficient (dMMR) CRC [30]. However, only 15% of CRCs are dMMR; therefore, the majority of CRC patients have tumours that are unresponsive to PD-1 inhibitor therapy. Immune therapies for mismatch repair proficient (pMMR) CRC could benefit from the addition of anti-MDSC therapies and our findings support exploring RASi as an immunomodulatory agent in this setting.

To our knowledge, this is the first study to demonstrate that the RASi captopril modulates the immune responses in the regenerating liver, in the absence of a tumour. We found that captopril significantly increased populations of CD8^+^ T cells, including T_RM_-like CD8^+^ T cells, in the regenerating liver in the absence of a tumour. This is in line with other studies that have also demonstrated that CD8^+^ T cells are key for hepatocyte proliferation and successful liver regeneration [31]. Conversely, the MDSC levels and subtype composition were not altered by captopril at this time point of liver regeneration, in the absence of a tumour. MDSC levels in the untreated controls were significantly higher in the combined tumour induction and liver regeneration group, compared with that seen in the absence of tumour. Furthermore, captopril treatment reduced MDSC levels in the combined tumour induction and liver regeneration group back to the baseline levels observed in the group with the absence of the tumour and liver regeneration.

Our study also demonstrated a crucial role for DN T cells in attaining effective antitumour immunity. The expansion of PD-1^+^ T_RM_-like DN T cells was unique to the CRLM model, suggestive of tumour specificity. This is in agreement with other studies that have demonstrated DN T cells possess important antitumour functions [32], and adoptive transfer of DN T cells is presently being explored as a novel cancer therapy [33].

In conclusion, this study showed that RASis reduced CRLM growth in the regenerating liver and altered immune cell composition by reducing populations of immunosuppressive MDSCs and enhancing populations of PD-1+ hepatic T_RM_-like lymphocytes. These results contribute to a growing body of literature suggesting that perioperative RASis may improve treatment outcomes for patients undergoing CRLM resection and should be explored further.

## 4. Materials and Methods

### 4.1. Animal Experiments

Animal experiments were approved by Austin Health Animal Ethics Committee (AEC-05435). Male CBA mice aged 10 weeks old and weighing 20–25 g were obtained from the Animal Resources Centre, Western Australia.

### 4.2. In Vivo Model and Cell Line

A mouse colorectal cancer cell line (MoCR), developed via dimethyl-hydrazine induction of colon carcinoma in CBA mice [34], was used for CRLM induction. Animals were divided into groups: with tumour induction and without tumour induction. The groups with tumour induction underwent surgical inoculation of CRLM via intrasplenic injection of 50,000 MoCR cells in 50 μL Ringer’s solution, followed by splenectomy (performed as one surgical procedure) (Figure 6) [35]. From Day 4, captopril (S-3 mercapto-2-methylpropionyl-L-proline, Sigma-Aldrich, St. Louis, Missouri, USA) 250 mg/kg/day or saline (control) was administered via intraperitoneal injection (Figure 6). On Day 7 following tumour induction, mice underwent a 70% partial hepatectomy, as previously described [35]. Carprofen (5 mg/kg) (Pfizer, New York, NY, USA) was administered for analgesia. Mice were culled on Day 16, and liver and tumour samples were retrieved for analysis.

The groups of mice not undergoing tumour induction followed the same treatment and hepatectomy timeline as outlined above but did not undergo tumour inoculation or splenectomy.

### 4.3. Stereometric Tumour Burden Analysis

Livers were fixed in 10% formalin for 24 h and stereometric tumour burden analysis was calculated, as previously described [13]. Briefly, livers were transected using a tissue fractionator. Liver slices were photographed and analysed using Image-Pro Plus. Tumour burden was calculated as a percentage of the whole liver volume.

### 4.4. Flow Cytometry

Tumour and liver samples were enzymatically digested (0.25 mg/mL liberase (Sigma) and 1 mg/mL DNAse (Sigma) in DMEM) at 37 °C for 40 min. Cell suspensions were filtered (100 μm cell filters, Corning), treated with red blood cell lysis buffer for 2 min at 37 °C, washed, and resuspended at 1 × 10^7^ cells/mL in cold FACS wash buffer (10% bovine serum albumin/5 mM EDTA/0.01% sodium azide in PBS pH 7). Briefly, 1 × 10^6^ cell aliquots were stained with either Cocktail 1: CD45-BV510, CD4-FITC, CD3-PE, CD44-PerCP, CD8-PECy7, PD-1-APC, CD69-APC-Cy7 (BD, Biosciences) and viability dye DAPI, Cocktail 2: CD45-BV510, CD103-FITC, CD3-PE, CD8-PECy7, PD-1-APC, CD4-APC-Cy7 (BD, Biosciences) and DAPI, or Cocktail 3: CD45-BV510, F4/80-FITC, Ly6C-PE, PD-L1-PECy7, CD11b-APC-Cy7 (BD, Biosciences), and DAPI. Samples were run on the Canto II (BD Biosciences) and analysed using FlowJo™ software.

### 4.5. Immunofluorescence

Following formalin fixation, livers were processed into paraffin blocks using standard techniques (Austin Pathology, Austin Health) and 4 μm sections were slide-mounted. Slides were dewaxed, and antigen retrieval (Tris buffer, pH 9) at 99 °C for 30 min was performed. Goat serum (10%) was used for blocking. Primary antibodies to PD-1 (rabbit monoclonal, Abcam, Cambridge, UK, 214421) and CD8 (rat monoclonal, Invitrogen, Waltham, MA, USA, 14-0808-82) were incubated overnight at 4 °C. Corresponding secondary antibodies were applied at 1:250 (Life Technologies AlexaFluor 488, Carlsbad, CA, USA, goat anti-rabbit IgG, A11034 and Life Technologies AlexaFluor 568, goat anti-rat IgG, A11077). DAPI 1:5000 was used for nuclear staining. Images were processed using confocal microscopy (Zeiss LSM780, Jena, Germany).

### 4.6. Statistical Analysis

GraphPad Prism was used for statistical analysis. Data were analysed using a two-tailed Student’s *t*-test. Graphical data are expressed as the mean +/− standard error of the mean.

## Figures and Tables

**Figure 1 ijms-23-05281-f001:**
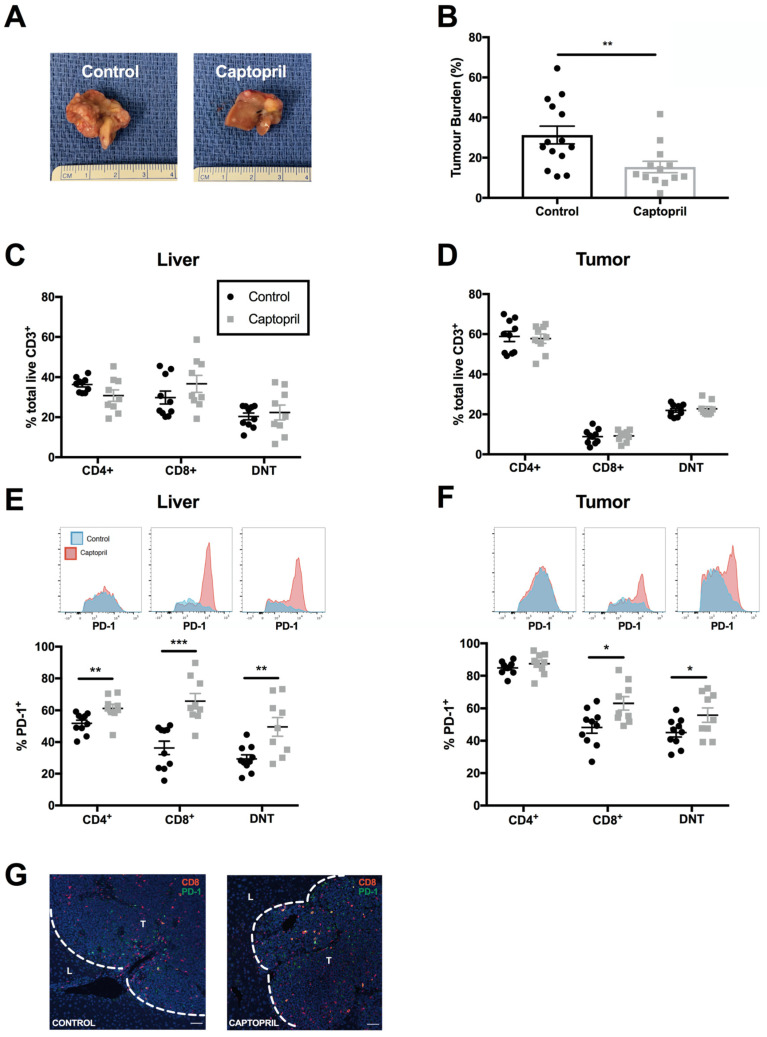
Captopril reduces tumour burden in regenerating liver and alters T-cell composition within liver and tumour tissues. Mice underwent surgical induction of colorectal liver metastasis via splenic injection, followed by splenectomy (Day 0). From Day 4 to Day 16, intraperitoneal injections of captopril 250 mg/kg/day or control (saline) were administered. On Day 7, mice underwent a 70% partial hepatectomy. Mice were culled on Day 16, and liver and tumour samples were retrieved for analysis: (**A**) representative photographs of liver and tumour appearance in the control and captopril treatment groups showed significantly greater tumour burden in the control group, compared with treatment; (**B**) tumour burden in the regenerating liver following partial hepatectomy in control and captopril-treated mice. Tumour burden = (tumour volume/total liver volume) × 100. Data compiled from two independently performed experiments are shown (control n = 14, captopril n = 13); (**C**,**D**) proportion of total CD3^+^ T lymphocytes that were CD4^+^, CD8^+^ and DNT (CD4^−^/CD8^−^) in liver (**C**) and tumour (**D**); (**E**,**F**) percentage of PD1^+^ T lymphocyte populations and insets of concatenated histograms of PD-1 fluorescence intensity on T lymphocyte populations in liver (**E**) and tumour (**F**). Data are compiled from two independently performed murine experiments (control n = 10 mice, captopril n = 9 mice). Significant differences were determined by an unpaired, two-tailed Student’s *t*-test. *p* < 0.05 (*), *p* < 0.01 (**), *p* < 0.001 (***); (**G**) representative immunofluorescence staining of CD8 (red) and PD-1 (green) in control and captopril-treated liver. Shown are tumour (T) and surrounding liver parenchyma (L).

**Figure 2 ijms-23-05281-f002:**
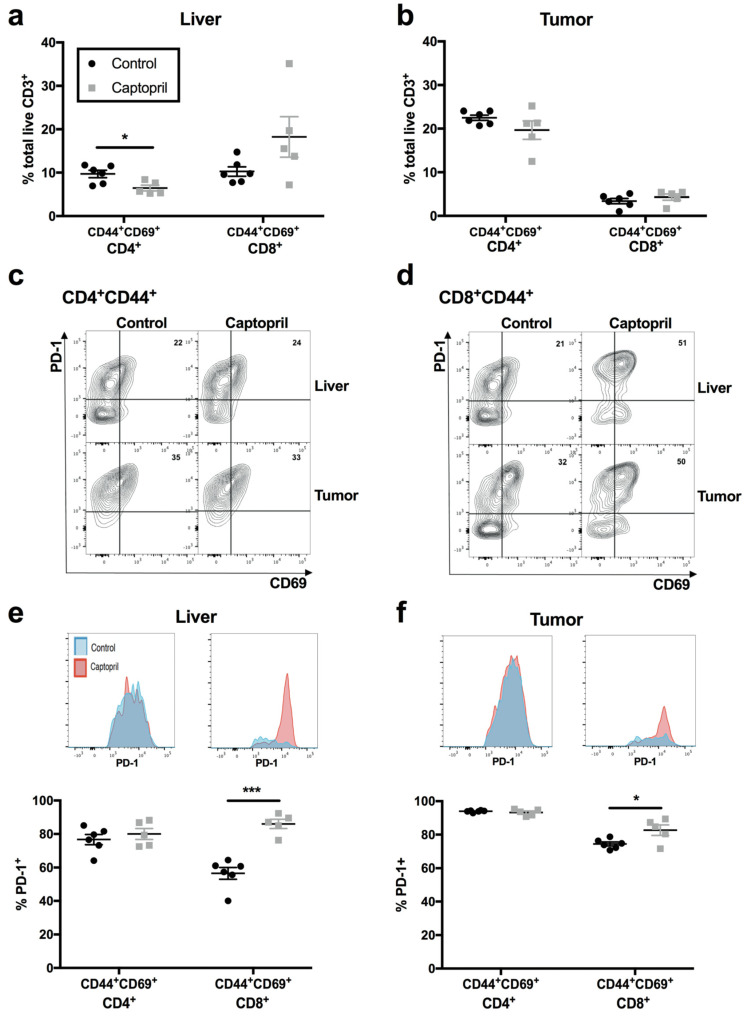
Captopril increases expression of the immune regulator PD-1 on tissue-resident memory (T_RM_)-like T cells in regenerating liver and tumour. Mice underwent surgical induction of colorectal liver metastasis via splenic injection (Day 0). From Day 4 to Day 16, intraperitoneal injections of captopril 250 mg/kg/day or saline (control) were administered. On Day 7, mice underwent a 70% partial hepatectomy. Mice were culled on Day 16, and liver and tumour specimens were retrieved for flow cytometry: (**a**,**b**) proportion of CD4^+^ or CD8^+^ T lymphocytes that were T_RM_-like (co-expressing CD44^+^/CD69^+^) in liver (**a**) and tumour (**b**); (**c**) concatenated contour plots demonstrating expression of CD69 and PD-1 on CD4^+^CD44^+^ T lymphocytes in liver (top panels) and tumour (bottom panels); (**d**) concatenated contour plots demonstrating expression of CD69 and PD-1 on CD8^+^CD44^+^ T lymphocytes in liver (top panels) and tumour (bottom panels); (**e**,**f**) percentage of PD-1^+^ T_RM_-like T lymphocyte populations and insets of concatenated histograms of PD-1 fluorescence intensity on T_RM_-like T lymphocyte populations in liver (**e**) and tumour (**f**). Data are representative of a single experiment (control n = 6 mice, captopril n = 5 mice). Significant differences were determined by an unpaired, two-tailed Student’s *t*-test. *p* < 0.05 (*), *p* < 0.001 (***).

**Figure 3 ijms-23-05281-f003:**
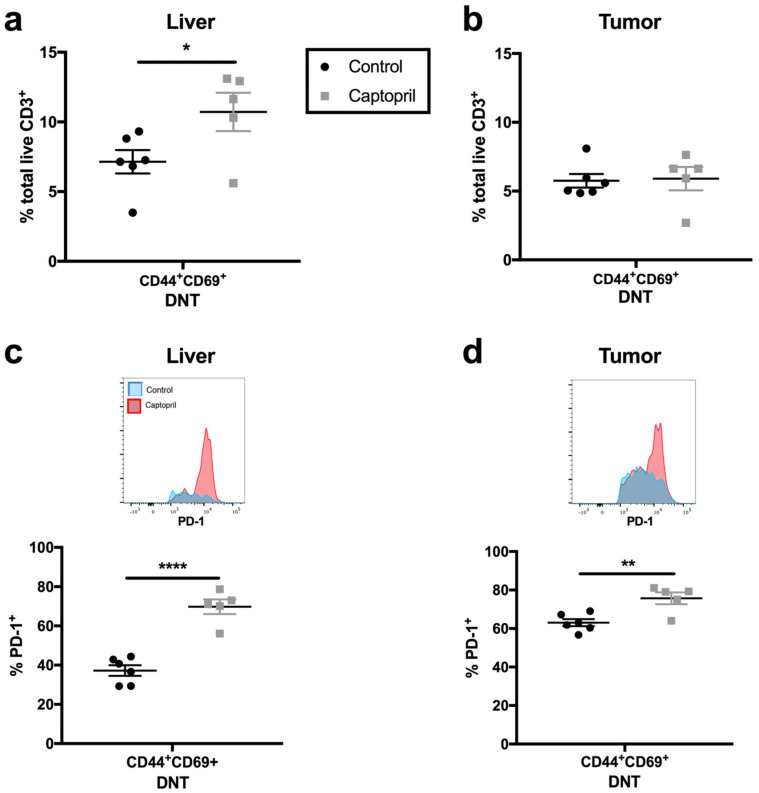
The renin–angiotensin system inhibitor (RASi) captopril increases the expression of the immune regulator PD-1 on tissue-resident memory (T_RM_)-like DNT cells in regenerating liver and tumour. Mice underwent colorectal liver metastasis tumour induction and were treated with either control (saline) or RASi captopril. From Day 4 to Day 16, intraperitoneal injections of captopril 250 mg/kg/day or saline (control) were administered. On Day 7, mice underwent a 70% partial hepatectomy. Mice were culled on Day 16, and liver and tumour specimens were retrieved for flow cytometry: (**a**,**b**) proportion of total CD3^+^ T lymphocytes that were DN T T_RM_-like (co-expressing CD44^+^/CD69^+^) in liver (**a**) and tumour (**b**); (**c**,**d**) percentage of PD-1^+^ T_RM_-like DN T lymphocytes and insets of concatenated histograms of PD-1 fluorescence intensity on T_RM_-like DN T lymphocytes in liver (**c**) and tumour (**d**). Data are representative of a single murine experiment (control n = 6 mice, captopril, n = 5 mice). Significant differences were determined by the unpaired, two-tailed Student’s *t*-test. *p* < 0.05 (*), *p* < 0.01 (**), *p* < 0.0001 (****).

**Figure 4 ijms-23-05281-f004:**
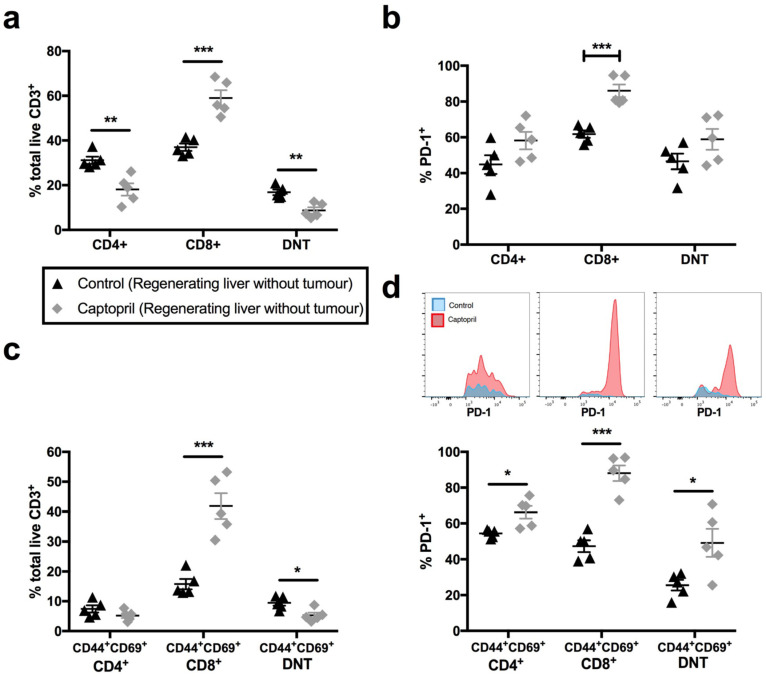
Captopril modulates T-lymphocyte populations in the regenerating liver, in the absence of tumour. Mice commenced treatment with either saline (control) or captopril 250 mg/kg/day and did not undergo tumour induction. Briefly, 70% hepatectomy was performed 3 days after the commencement of treatment, and mice were culled nine days following partial hepatectomy in keeping with prior experiments. Liver specimens were retrieved for flow cytometry: (**a**) proportion of total CD3^+^ T lymphocytes that were CD4^+^, CD8^+^ and DN T (CD4^-^/CD8^-^); (**b**) percentage of PD-1^+^ T lymphocyte subpopulations; (**c**) proportion of CD4^+^, CD8^+^ and DN T lymphocytes co-expressing T_RM_ markers, CD44^+^ and CD69^+^; (**d**) percentage of PD-1^+^ T_RM_-like T lymphocyte populations and insets of concatenated histograms demonstrating fluorescence intensity of PD-1 on T_RM_-like populations. Data are representative of a single experiment (control, n = 5 mice, captopril, n = 5 mice). Significant differences were determined by an unpaired, two-tailed Student’s *t*-test. *p* < 0.05 (*), *p* < 0.01 (**), *p* < 0.001 (***).

**Figure 5 ijms-23-05281-f005:**
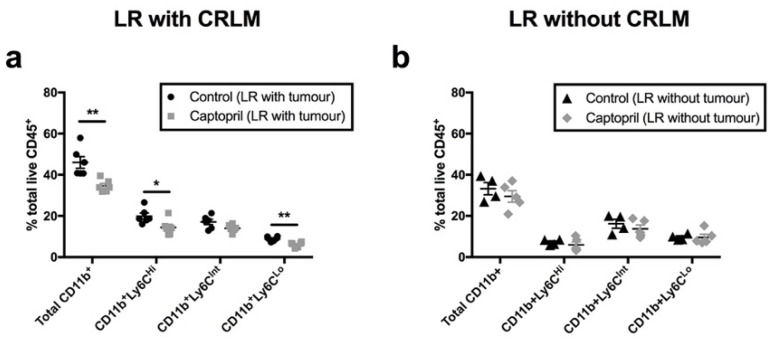
The renin–angiotensin system inhibitor (RASi) captopril reduces CD11b^+^ myeloid-derived suppressor cell populations within the regenerating liver (LR) following tumour induction, down to baseline levels observed in regenerating liver without tumour. One group of mice underwent surgical induction of colorectal liver metastasis via splenic injection (Day 0) and one group of mice did not undergo tumour induction. From Day 4 to Day 16, intraperitoneal injections of captopril 250 mg/kg/day or saline (control) were administered to all mice in both groups. On Day 7, all mice underwent a 70% partial hepatectomy. All mice were culled on Day 16, and liver and tumour specimens were retrieved for flow cytometry: (**a**,**b**) proportion of total CD45^+^ myeloid cells that were CD11b^+^, CD11b^+^Lyc6C^Hi^, CD11b^+^Ly6C^Int^, and CD11b^+^Ly6C^Lo^ in livers of mice following tumour induction and partial hepatectomy (**a**) and in livers of mice following partial hepatectomy alone (no tumour) (**b**). Data are representative of two independently performed murine experiments (control n = 6 mice, captopril n = 6 mice). Significant differences were determined by an unpaired, two-tailed Student’s *t*-test. *p* < 0.05 (*), *p* < 0.01 (**).

**Figure 6 ijms-23-05281-f006:**
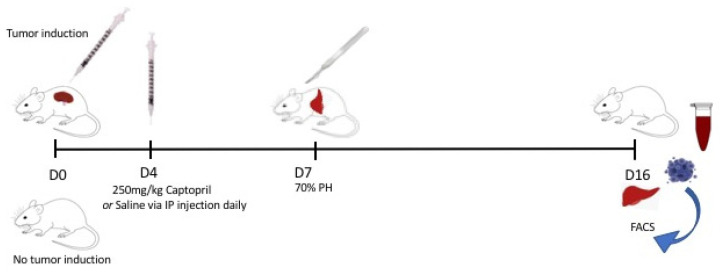
Experimental methodology. Mice were allocated to either tumour induction or no tumour induction groups, and this was performed on Day 0. From Day 4 to Day 16, half the mice in each group received treatment with intraperitoneal injections of captopril 250 mg/kg/day and the other half received intraperitoneal injections of saline (control). On Day 7, all mice underwent a 70% partial hepatectomy. Mice were culled on Day 16, and liver and tumour (if applicable) samples were retrieved for analysis.

**Table 1 ijms-23-05281-t001:** Ratios of total CD4^+^:CD8^+^ lymphocytes with their respective ratios for CD44^+^CD69^+^ and CD44^+^CD69^+^PD-1^+^ populations in liver.

CD4:CD8 Ratios	Total	CD44^+^CD69^+^	CD44^+^CD69^+^PD-1^+^
Treatment—Captopril (mean ± SD)	0.91 ± 0.36)	0.45 ± 0.25	0.44 ± 0.27
Control—Saline (mean ± SD)	1.57 ± 0.35	1.00 ± 0.33	1.33 ± 0.31
*p* value	**0.01**	**0.01**	**0.0007**

Significant differences were determined by the unpaired, two-tailed student’s *t*-test. Significant *p* values are in bold. SD—standard deviation.

## Data Availability

The data presented in this study are available on request from the corresponding author.

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
