# Peer review of "Captopril, a Renin–Angiotensin System Inhibitor, Attenuates Tumour Progression in the Regenerating Liver Following Partial Hepatectomy"

_ijms, 2022, doi:10.3390/ijms23095281_

Round 1
Reviewer 2 Report
In the present manuscript submitted by Georgina E. Riddiough et al, the autors aimed to explore whether RASi attenuate tumor progression in the regenerating liver following partial hepatectomy, which seems a bit interest. But, a generally superficial and under-developed design of the main theme of manuscript made me confused that there may be a simply combination of their previous work, and the content is not going deep and novel enough.
The anti-tumor effect that Captopril exerted and the immunomodulatory effects of renin–angiotensin system inhibitors on T lymphocytes in mice
have been stated in a certain extent in their earlier work (PMID: 34073112, Captopril, a Renin-Angiotensin System Inhibitor, Attenuates Features of Tumor Invasion and Down-Regulates C-Myc Expression in a Mouse Model of Colorectal Cancer Liver Metastasis. PMID: 32448803, Immunomodulatory effects of renin–angiotensin system inhibitors on T lymphocytes in mice with colorectal liver metastases). So that, the content of this manuscript would seem be a certain cross-talk between their two published paper (PMID: 34073112, PMID: 32448803).
Round 2
Reviewer 1 Report
The authors have clarified all of my major concerns. I recommend this work to be published in IJSM.
Author Response
We would like to thanks Reviewer #1.
Reviewer 2 Report
The content of this manuscript is not innovative and significant enough, and the presentation is not highly solid and proper to exert the effect of attenuating tumor progression.
Author Response
We thank the Reviewer for their feedback. As previously stated, we have amended the Introduction to more clearly state that this work extends our previous publications and is original. This was the first publication to examine the role of the TRM and myeloid cells within the regenerating liver post partial hepatectomy for liver metastasis.
A second major strength of our work is that it utilise a homogenous mouse tumour model which closely mimics the clinical scenario, this is advantageous over other mouse models such as xenograft models and differs from earlier work.
We would appreciate it if the reviewer could please provide more feedback about why they believe this work is not “highly solid and proper” so that we can more specifically address their concerns.
Please see the attachment.
